# Determining the Key Education Priorities Related to Heart Failure Care in Nursing Homes: A Modified Delphi Approach

**DOI:** 10.3390/healthcare12151546

**Published:** 2024-08-05

**Authors:** James McMahon, David R. Thompson, Christine Brown Wilson, Loreena Hill, Paul Tierney, Jan Cameron, Doris S. F. Yu, Debra K. Moser, Karen Spilsbury, Nittaya Srisuk, Jos M. G. A. Schols, Mariëlle van der Velden-Daamen, Gary Mitchell

**Affiliations:** 1School of Nursing and Midwifery, Queen’s University Belfast, Belfast BT9 7BL, UK; david.thompson@qub.ac.uk (D.R.T.); c.brownwilson@qub.ac.uk (C.B.W.); l.hill@qub.ac.uk (L.H.); p.tierney@qub.ac.uk (P.T.); gary.mitchell@qub.ac.uk (G.M.); 2School of Clinical Sciences at Monash Health, Monash University, Melbourne, VIC 3168, Australia; jan.cameron@monash.edu; 3School of Nursing, University of Hong Kong, Hong Kong, China; dyu1@hku.hk; 4College of Nursing, University of Kentucky, Lexington, KY 40536, USA; debra.moser@uky.edu; 5School of Healthcare, University of Leeds, Leeds LS2 9DA, UK; k.spilsbury@leeds.ac.uk; 6Faculty of Nursing, Rajamangala University of Technology, Thanyaburi, Pathum Thani 12110, Thailand; nittaya_sr@rmutt.ac.th; 7Department of Health Services Research and Department of Family Medicine, Care and Public Health Research Institute (CAPHRI), Maastricht University, P.O. Box 616, 6200 Maastricht, The Netherlands; jos.schols@maastrichtuniversity.nl (J.M.G.A.S.); m.vandervelden@maastrichtuniversity.nl (M.v.d.V.-D.)

**Keywords:** heart failure, nursing homes, older people, Delphi technique, education, health

## Abstract

There is currently a limited understanding of what nurses in nursing homes view as the key education priorities to support their ability to provide the appropriate care for residents with heart failure (HF). A modified Delphi technique was utilized to gain a consensus on the key education priorities for nurses working in nursing homes in Northern Ireland. An initial list of items (*n* = 58), across 19 domains, was generated using the findings of a scoping review and stakeholder interviews, and a review of available clinical guidelines. Two rounds of surveys were undertaken. Items were presented using a 5-point Likert scale, with an additional exercise in the second round to rank the domains in order of importance. Fifty-four participants completed the first-round survey and 34 (63%) returned to complete the second. The findings highlight the importance of providing nurses in nursing home settings with general HF education and the delivery of person-centered care. Participants perceived education around technology for the management of HF and quality improvement or research methodologies associated with HF in nursing homes as lower priorities. This study illuminates key priorities from nursing home nurses regarding HF education that are applicable to this care setting.

## 1. Introduction

Heart failure (HF) is a clinical syndrome characterized by structural and/or functional changes to the heart [1]. The global prevalence of HF is estimated to be 64.3 million cases [2]. It is anticipated that this number will continue to rise due to an ageing population, improved diagnostic testing, and improved rates of survival following an acute cardiovascular event [2,3]. The prevalence of HF increases with age, rising from 1% in those <55 years to 10% in those >70 years [4], with a mean age of 75.2 years [5]. As a result, it is no surprise that HF prevalence is high among nursing home residents, with rates between 10–45% reported [6,7,8,9,10].

Registered nurses working in United Kingdom (U.K.) nursing homes reported that they require a particular set of skills, knowledge, competencies, and experience to provide high-quality care for older residents [11]. Nursing home staff contribute to the care of those with HF, yet often lack the knowledge and competency to provide appropriate care to those living with the condition [8,12]. The lack of knowledge and competency, combined with the more complex needs of residents with HF [13], highlights the importance of providing staff education to enhance care provision [12]. Improvements in nurses’ knowledge have the potential to benefit patient care, improve staff job satisfaction, and reduce both warranted and unnecessary acute care utilization [14].

Qualitative research with nurses working in a nursing home setting has identified key themes related to HF care on which further training was required [8]. This includes improving knowledge and skills associated with identifying HF signs and symptoms, the purpose of daily weighing, indicators of worsening HF status, purpose of a sodium restricted diet, and materials to improve the residents’ understanding of HF [8]. Such findings highlight the importance of nursing home nurses to possess the knowledge and competence to provide maximum support to residents with HF to optimize their care and quality of life. A recent scoping review of educational interventions in nursing homes for optimizing care provision for residents with HF reported improvements in staff knowledge, as well as greater confidence and self-efficacy toward providing care [15]. However, due to a lack of process evaluations of the interventions, it is difficult to ascertain what nurses, as key stakeholders, view as the key education priorities that will best support their ability to provide appropriate HF care. Thus, it is necessary to uncover the key education priorities of nurses working within the nursing home setting for optimizing care provision to residents of nursing homes with HF. 

The aim of this study was to gain a consensus on the key education priorities associated with providing care to those with HF in nursing homes and enable the development of a core outcome set (COS). This was achieved through two objectives:To compile a list of outcomes identified as key education priorities from a scoping review, stakeholder interviews, and a review of available clinical guidelines.To condense these outcomes into a COS using a Delphi questionnaire completed by a panel of experts.

## 2. Materials and Methods

### 2.1. Design

A COS can be defined as a set of agreed outcomes applicable to all trials in a particular area of health [16]. Whilst not the only outcomes that should be reported in trials, a COS can establish a minimum outcome set that should be considered [17]. The Delphi method is an iterative, structured, multistage approach, with each stage (or round) building on the results of the previous one [18,19]. Allowing for the gathering of information across diverse geographical locations, the Delphi method has been advocated as a method of reaching a consensus for a COS within the health and social sciences [18,20,21]. This study employed a modified Delphi approach delivered through a series of online surveys. A modified Delphi is similar to a traditional Delphi approach in the procedure and intended outcome. The key difference is in the first round of surveys, with the traditional Delphi providing an open-ended questionnaire to develop initial COS items, whereas these items are pre-selected for the first round in the modified approach [22]. As a result, a modified Delphi typically requires two rounds to gain a consensus on all items [23]. 

Ethical approval for this study was granted by the Faculty of Medicine and Health Sciences, Queen’s University Belfast (MHLS 23_114). This study is part of a larger research project that seeks to develop a digital intervention aimed at optimizing the quality of life of nursing home residents with HF [24].

### 2.2. Participant Recruitment to Delphi Panel

Registered nurses working within the nursing home setting in Northern Ireland (NI) were recruited to the Delphi panel in this study. No eligibility criteria were set for years of experience working within the nursing home setting. It was deemed important to gather the views and opinions of nurses from all levels of experience to ensure a holistic view of key education priorities within this setting were gathered. As the results of the Delphi would guide the development of an intervention for nurses working within a nursing home setting only, it was determined that the views of other stakeholders, such as residents or their family members, would not be required. It was determined that the perspectives of others outside of nurses working within nursing homes may be too heterogeneous to be combined into a COS targeted at this specific population. Participants were recruited through the ‘Queen’s University Belfast Care Home Research Network’. This group represented more than 100 nursing homes across NI. Additional recruitment was carried out using professional networks established by the research team. 

The established ‘Queen’s University Belfast Care Home Research Network’ made it possible to recruit participants from ‘research ready’ nursing homes, previously proven as an effective method for recruitment within this setting [25]. Nursing home managers often act as the gatekeeper between the researcher and staff [26]. Contacting staff in nursing homes directly can prove problematic due to limited communication pathways, including staff having a lack of access to work computers and email [27,28,29]. Further, as the managers of these nursing homes were committed to engaging in research, a less intense and resource-heavy approach toward recruitment was required. As such, email was considered as an appropriate method of recruitment. 

No sample size guidelines currently exist for a Delphi panel [23,30], with samples between 13–22 reported [21]. Typically, Delphi panels are comprised of 10–100 participants, with approximately 50 panel members deemed optimal for a Delphi with a homogenous sample [23]. Thus, it was determined that 100 participants would be invited to participate, anticipating a 50–75% response rate whilst considering potential attrition between rounds. An invitation letter and information sheet were delivered via email to the nursing home managers of 100 nursing homes across NI. Managers were eligible to take part if they were registered nurses, whilst also being requested to distribute the email with the registered nurses employed in their nursing home. If interested, participants were required to contact the research team via email expressing an interest and were then provided with a URL link to the first online survey. Participants were required to complete an online consent form to access to the survey, with access to the survey denied if they did not consent to participate. 

### 2.3. Development of Delphi Survey

The list of items generated for the Delphi survey were based on two pieces of empirical research, a scoping review of educational interventions for optimizing care provision to those with HF in nursing homes [15], and one-to-one interviews (*n* = 16) with nurses working within the nursing home setting (manuscript in preparation). Additionally, three pieces of clinical guidance were used to guide the development of the initial items; the standards of education and practice for nurses working in nursing homes [31]; a curriculum for specialist HF nurses [32] and; a competency framework for HF specialist nurses [33]. Using the guidance and empirical research, an initial ‘long list’ of items was compiled by J.M. and G.M. The ‘long list’ of items was reviewed by L.H., a HF specialist nurse, who assisted in refining these into an appropriate and valid list. All members of the research team had the opportunity to review and provide feedback on the refined list prior to progressing to round one, with a total of 58 items included (Appendix A). Items were presented across 19 domains associated to HF care for both the general population and older people in nursing homes, e.g., ‘General Heart Failure Education’, ‘Person-Centred Care for Heart Failure Residents’, ‘Symptom Management and Palliative Care’, and ‘Interprofessional Communication’. 

### 2.4. The Delphi Process

Each round, participants were required to provide their perspective on the level of priority that each item held in relation to optimizing care for residents with HF. A 5-point Likert-type scale, preferred by participants in Delphi studies [34], was utilized (1—not a priority, 2—low priority, 3—medium priority, 4—high priority, 5—essential). As no guidelines currently exist for the number of rounds required to reach a consensus, 2–3 rounds are typically conducted [23,35,36]. Guided by prior research, a flexible approach was applied with no fixed number of rounds set [37]. Rather, criteria were set to determine whether an item would be included in the final COS. The Delphi study ceased once consensus was achieved for all items [23], classified as either:Consensus in—75% voting ≥4 (e.g., high/essential-priority items) and less than 15% ≤2 (e.g., low/non-priority items)Consensus out—75% voting ≤2 (e.g., low/non-priority items) and less than 15% ≥4 (e.g., high/essential-priority items)orNo consensus achieved—does not meet inclusion or exclusion criteria in successive rounds.

Each round of the Delphi survey remained open for completion for three before being closed for analysis. A reminder email to complete the survey was sent to all participants one week prior to the close of each round. 

#### 2.4.1. Delphi Round One

Participants were provided with the URL to the first-round survey, alongside instructions on how to complete the online form via email. The survey was comprised of all 58 items generated from the empirical research and clinical guidance. Demographic data were also collected during the first round, including age, gender, ethnicity, and years worked in the nursing home setting. The first-round survey was anticipated to take a maximum of ten minutes to complete. 

#### 2.4.2. Delphi Round Two

Participants were contacted three weeks after closing the first-round survey by email with a URL link to the second-round survey, comprised of two sections. Section one presented only those items from the first survey for which no consensus had been reached (‘no consensus achieved’). Group responses for items in which no consensus had been reached during round one was provided to participants in the form of graphs with frequencies. Providing group responses facilitates reflection and aids participants in reconsidering their initial responses [38]. In section two, the ten domains with the most items classified as ‘consensus in’ after round one were presented to the participants. Participants were required to subjectively rank each domain in importance from 1 to 10 (1 being the most important and 10 being the least important). The second-round survey was anticipated to take a maximum of five minutes to complete.

### 2.5. Analysis of Delphi

All data from each round were imported and analyzed using SPSS statistics package v29. Descriptive statistics were used to analyze and present participant demographic data. Group responses for each item were analyzed to determine frequency of responses, and the mean, median, and mode for each item (Appendix A). First-round survey data were analyzed before progressing to the second, in line with the consensus criteria. 

## 3. Results

### 3.1. Participants

Fifty-four participants completed the first-round survey, with 34 (63%) returning to complete the second (Figure 1). Mean age of participants was 28.7 years (SD = 9.2). Most participants were female (91%) and white (90%). The years working within the nursing home sector varied greatly, with most working either one year (37%) or more than 10 years (20%). 

### 3.2. Delphi Round One

Of the 58 items, 39 (67%) were deemed to be of high priority or essential (≥4) and would be included in the final COS. No items were deemed to have been of low or no priority (≤2), with the remaining 19 items meeting neither the inclusion nor exclusion criteria. Three domains had no items for which consensus was met, (i) Technological Advancements in Heart Failure Management, (ii) Quality Improvement and Evidence-Based Practice, and (iii) Education on Research and Innovation. 

### 3.3. Delphi Round Two

Of the 19 items in which no consensus had been achieved during round one, three met the ‘consensus in’ criteria, with 42 items included in the final COS. The remaining 16 items achieved ‘no consensus’ in consecutive rounds and therefore would not be included in the final COS. After two rounds, two domains remained with no items meeting the consensus criteria and thus would not be represented in the final COS, (i) Technological Advancements in Heart Failure Management, and (ii) Education on Research and Innovation. As a consensus had been agreed on all items following round two, the Delphi study was considered complete at this stage. The top ten domains ranked from most important (1) to least important (10) (Table 1) are outlined below.

## 4. Discussion

This is the first study to determine the key education priorities of nurses for optimizing care of people with HF in nursing homes. The findings emphasize the importance of providing nurses working within the nursing home setting with education around HF and the delivery of person-centered care to enhance care provision to residents with HF. Ranking as the number one priority in the domains exercise, nursing home staff in this study further support the existing evidence that knowledge around HF in nursing homes is currently lacking [8,12] and that the provision of education around the condition is undoubtedly required. Such education would include pathophysiology, types and signs of HF, clinical manifestations, recognition and management of exacerbations, and education surrounding assessment tools and diagnostic tests for HF. 

Person-centered care for residents with HF was ranked as the second highest priority in this study, similar to a prior Delphi study aimed at establishing a set of research priorities in U.K. nursing homes [39]. The authors postulate that person-centered care was viewed as a high priority due to UK policy developments for care services already, indicating this as a key aspect of quality care within care services [39]. Being viewed as the basis for quality of care in nursing homes, it is unsurprising that person-centered care for residents with HF was viewed as a high priority. Items associated with person-centered care included the promotion of enhanced communication skills for improved engagement and support, and techniques for assessing and addressing emotional and psychosocial aspects of HF, ranked highly in the current study. This suggests that irrespective of the chronic condition a nursing home resident may be living with, nursing home staff strive to actively engage the residents in their own care. 

Of interest is the differentiation between end-of-life care (EOLC), provided in the final weeks or days of someone’s life, and palliative care/symptom management, provided in the years/months before EOLC. Palliative care and symptom management was ranked as a higher priority than EOLC and bereavement, likely due to HF being a chronic condition. However, the results may also suggest that nursing home staff currently possess the competence to provide EOLC to those with HF, as well as bereavement support to other residents and family members, yet lack the confidence surrounding symptom management and palliative care associated with a progressive chronic condition such as HF. Issues surrounding a lack of knowledge around the practices and principles of general palliative care among nursing home staff have been reported previously [40,41,42]. Additionally, the medications, symptoms, and emergency triad (4, 5, and 6) that emerged from the ranking exercise suggest that nursing home staff recognize the importance of general medical care in caring for someone living with a chronic condition, i.e., medications to stay well, management of symptoms, and how to prepare for a rapid decline in HF status of a resident. 

Two domains, ‘Technological Advancements in Heart Failure Management’ and ‘Education on Research and Innovation’, were not represented in the final COS. Items associated with technological advancements focused on updating staff on methods for monitoring changes to HF status and education on the role of telemedicine for remote monitoring. Telemedicine ranking as a low priority was surprising, with the utilization of telemedicine increasing dramatically as a result of the COVID-19 pandemic, estimated to have grown from 13–39% pre-pandemic [43] to 84% post-pandemic [44] in the United States of America (USA). Further, a study conducted in NI to determine the capacity and enthusiasm for telemedicine implementation for delivering psychiatric treatment to nursing home residents reported a strong interest by nursing home staff, with approximately 70% of the participating nursing homes currently possessing the appropriate facilities and equipment necessary for implementation [45]. Prior research has reported that nursing home staff, with no prior experience of telemedicine, view the approach as dehumanizing and shifts the focus away from person-centered care. However, long-term follow-up reported that these concerns were fallacious, with staff agreeing on telemedicine’s potential for optimizing quality of care [46]. Thus, it is possible that participants in the current study have little to no experience of telemedicine and perceive this approach as potentially impacting their ability to provide person-centered care, a factor viewed as a high priority. Further, an integrative review to evaluate and appraise telemedicine and telehealth in nursing homes reported additional barriers toward implementation and uptake by staff, including high staff turnover leading to training issues, difficulties for staff coping with change, and concerns over inadequate allocation of staff time toward implementation [47]. Additionally, the number of years working within the nursing home setting, ranging from 1–10+ years, may have influenced participants’ responses. It is likely that those with less experience are younger nurses and may be more open to the implementation of telemedicine compared to their more experienced and older counterparts.

The domain of ‘education on research and innovation’ included items around informing nursing home staff of ongoing HF research and education to encourage staff to participate in research associated with HF in nursing homes. Engaging nursing home staff is historically challenging, with staff citing a lack of knowledge of research and the associated processes, time constraints, changing shift patterns, and high staff turnover as key barriers [14,26,48,49,50]. Additionally, staff indicate that clear management by nursing home managers would be required if staff were to participate in research to help offset the potentially higher workload and strain associated with participation but have concerns that this may not be provided [49]. Such concerns may be warranted, with nursing home managers reportedly having negative attitudes toward research, and therefore may not provide the necessary management of staff required to successfully foster engagement by staff [44]. Although the COVID-19 pandemic has indirectly had a positive impact on public awareness of research [51,52], research education was viewed as a non-priority by the participants in our study, suggesting that further exploration is required to improve awareness and engagement with research by nursing home staff. To assist uptake in research within the nursing home setting, a series of questions have been developed to support nursing home managers in determining the capacity, readiness, and relationships of their home to support a research trial through discussions with staff, residents, and their family and friends [53].

There are several limitations of this Delphi study to consider. First, only registered nurses working in nursing homes in NI were recruited. Although this approach is in line with recommendations for conducting a Delphi study with homogenous samples, the generalizability of the findings may be limited as the educational priorities identified may differ to staff in nursing homes from different professions (i.e., care assistants). Nonetheless, the participants possessed a wide range of experience as a nurse within the nursing home setting, providing greater perspective for each item and enhancing the validity of the findings. Second, participant attrition, a common issue of the Delphi method [54], was relatively high with only 63% of participants completing both rounds. Higher rates of attrition have the potential to impact the consensus results. However, due to utilizing an online approach in this study, risk of attrition is likely to have been minimized when compared to employing a paper-based approach [55,56]. Future research would benefit from determining the reasons for participant attrition and to identify methods of improving adherence to Delphi studies with this population. Further, as there were minimal changes in consensus ratings of items between the first and second round, the impact of attrition, if any, is likely minimal. 

## 5. Conclusions

In conclusion, this study has identified the key education priorities related to the care of residents living with HF in nursing homes in NI. With HF education being classified as the top education priority in this Delphi study, our findings appear to support that of prior research, being that nurses working within the nursing home setting lack the education and understanding of HF. This in turn may impact their competency in providing care and assisting in the management of the condition for residents with HF. Thus, it is imperative that education is provided to optimize the quality of care provided and improve outcomes of nursing home residents with HF. However, the delivery of further training and continuing professional development (CPD) in nursing homes is challenging due to a lack of staff cover limiting available free time, and staff often having to attend courses outside of working hours unpaid [11]. Nursing home staff have indicated a preference for further training to be delivered whilst on the job, particularly relating to the management of long-term conditions [11]. Thus, the delivery of a digital education intervention, for which the findings of our study will inform the development of, may meet the demands of nursing home nurses and subsequently improve the likelihood of success.

## Figures and Tables

**Figure 1 healthcare-12-01546-f001:**
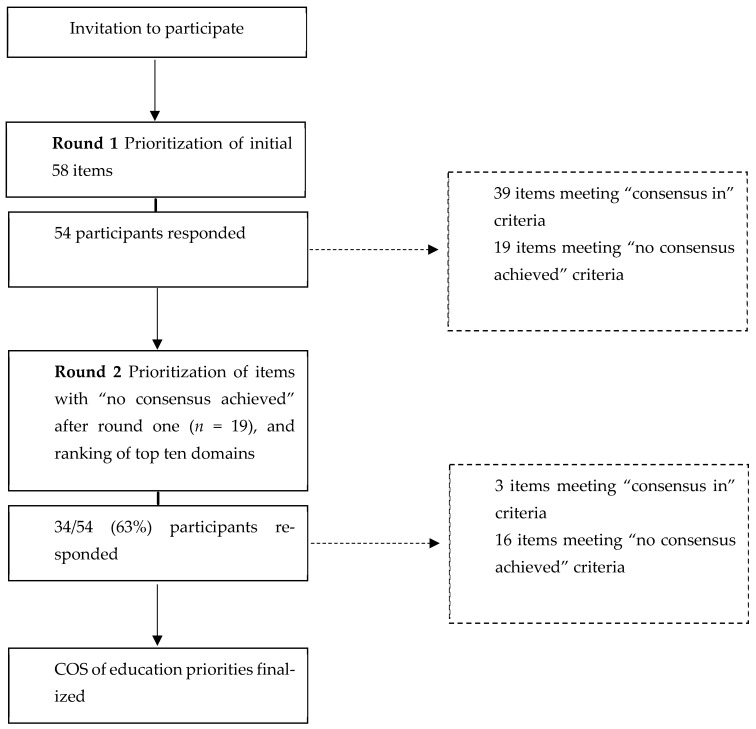
Delphi study flowchart.

**Table 1 healthcare-12-01546-t001:** Top ten domains ranked in order of priority (highest to lowest priority in descending order).

Rank	Category
1	General Heart Failure Education
2	Person-Centred Care for Heart Failure Residents
3	Multidisciplinary Collaboration
4	Medication Management and Adherence
5	Emergency Preparedness and Crisis Management
6	Symptom Management and Palliative Care
7	Nutrition and Fluid Management
8	Falls Prevention and Safety
9	End-of-Life Care and Bereavement Support
10	Ethical and Legal Considerations in Heart Failure Care

## Data Availability

The original contributions presented in the study are included in the article/Appendix A; further inquiries can be directed to the corresponding author/s.

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
