# Peer review of "Determining the Key Education Priorities Related to Heart Failure Care in Nursing Homes: A Modified Delphi Approach"

_healthcare, 2024, doi:10.3390/healthcare12151546_

Round 1

Reviewer 1 Report

Comments and Suggestions for Authors

The authors used a validated and formal method (modified Delphi method) to identify consensus regarding the care priorities of the management of the patient with heart failure. To this end, 100 expert nurses (nursing home managers) chosen from the more than one hundred nursing homes belonging to the Queen's University Belfast Care Home Research Network were contacted". Among the one hundred health workers initially contacted, only 34 completed the two Delphi rounds (34%).  The healthcare professionals involved had to select 19 care domains built on a total of 58 items. The Delphi method was implemented with only two rounds; consensus was reached on 42/58 items (72.4%) relevant to 10/52 domains (52.6%; those indicated in table 1).

The essential strength of the research is represented by having used a validated method as a tool to reach consensus among experts.

The research has several limitations, some of which have been highlighted by the authors themselves. These limits do not affect the possibility of publishing the article but in my opinion require some clarifications and some additions.

First The Delphi method requires that the experts involved do not know each other, to avoid being influenced in the  choices by the opinions of others received informally. The authors instead declare that they have invited the nursing home managers to involve other nurses working in the same nursing home in participating in the research. The authors should specify in the article how many of the 'responder' operators worked in the same structure.

Second. The first round of the Delphi method was organized by sending participants a series of items and domains identified by the project group through literature sources (bibliographic articles, guidelines) and through one-to-one interviews with nurses operating in the same nursing home settings. The authors specify 'manuscript in preparation' but should anticipate in the article at least the criteria with which these interviews were planned and how many nurses were involved

Third.The number of responders (34/100) is very low (the problem linked to the representativeness of the sample was discussed by the authors themselves). This is not surprising, given that the recruitment of respondents was carried out exclusively via email. Investigations of this type should be carried out by implementing two or three reminder cycles (e.g. telephone reminders), possibly also investigating the causes of non-adherence to the initiative. Authors should specify the reasons why this was not done (e.g. lack of dedicated staff, limited time available, etc.)

Fourth. The number of rounds - frankly - seems also too little to me given that consensus was not reached in 16/58 items. The authors should at least mention the rationale underlying this choice (e.g. logistical difficulties in contacting interviewees or linked to the need to produce results in a reasonable time, etc.)

Fifth. One point is not clear to me (line 155). In the second stage, respondents received material classified into two sections. The first section concerned the issues on which consensus had not been reached, accompanied by tables and graphs and aimed at inviting interviewees to anonymously compare their opinions with those of others. The second section contained the name of ten domains characterized by the greatest number of items for which consensus had been reached. Up to this point everything is OK. The interviewee had the task of reclassifying these ten domains according to a rank of importance based on his own opinion; the project team would then use these scores for the final reclassification of the 10 domains into a new priority rank: is that right?

Reviewer 2 Report

Comments and Suggestions for Authors

The authors report the results of a survey of nursing professionals in nursing homes. The aim was to demonstrate the need and importance of increasing training in heart failure among these health care professionals.  However, the results are based on the subjective impression of these professionals regarding their educational needs. With the results obtained, the authors conclude that "nurses working within the nursing home setting currently lacking an understanding of heart failure," a conclusion that is probably not supported by the survey results. An assessment of nurses' knowledge of the care of patients with heart failure might have corroborated the authors' statement in their conclusions.

On the other hand, the small study population, the high dropout rate, and the local setting of the study subjects make it very difficult to generalize the data, making the results irrelevant. The authors themselves rightly list these problems among the "limitations" of the study. However, in my opinion, these problems are too many to recommend publication of the study.

Finally, I believe that further explanation of the design and the various items that make up the survey is needed.

Reviewer 3 Report

Comments and Suggestions for Authors

There is a pressing need for a deeper understanding of the key education priorities perceived by nurses in nursing homes to enhance their capacity to deliver optimal care to residents with heart failure (HF). The authors of this study have employed a modified Delphi technique to establish a consensus on the crucial education priorities for nurses in nursing homes in Northern Ireland. Their work sheds light on a vital issue in nursing education for individuals with heart failure, and the reviewers acknowledge the significance of their findings. However, the authors still need to address the following issues.

Major issues:

1. The study design section(line 73) is currently heavy on conceptual content, such as COS and Delphi methods, which may make it challenging for readers to understand the study's design. It is suggested that The authors reorganize this section into a more reader-friendly format, with additional details about the study's design. Furthermore, the requirements for the experts when completing the questionnaire should be clearly outlined.

2. Participant recruitment is a crucial step in the Delphi method, and the professional experience of the experts is paramount for providing accurate responses to the questionnaire. However, the authors needed to clearly define the scope of the experts, such as the minimum number of years of experience in the nursing profession. It is recommended that the authors include this information and also consider including specialized physicians working in heart failure treatment in the expert panel.

3. the first ten areas in the prioritization contain essential information, i.e., they are prioritized from high to low, implying that each area has a different weighting; how this ranking is interpreted to be reflected in the discussion section.

4. in the results section (line 171), "The years working within the nursing home sector varied greatly, with most working either one year or more than ten years. " This confirms my concern that the number of years a specialist has worked in their profession can greatly impact the questionnaire results. For example, in 'Technological advances in heart failure management,' younger practitioners may be more interested in telemedicine, but senior practitioners with more experience may find it dehumanizing.

5. In the limitations section(line271), the authors suggest that one of the limitations is the relatively small sample size, which may be an inaccurate conclusion, as 54 experts participated in the first round and 34 in the second round, which is a sufficient number of experts to satisfy the requirements of a Delphi survey, and more importantly, the representativeness of the experts who participated.

Minor issues:

1. Please add the time of completion of the first round of survey and the second round of survey.

2. line 164, " Group responses for each item were analyzed to determine the frequency of responses, and the mean, median and mode for each item. "It is hoped that the appropriate data will be added to the results section.

Round 2

Reviewer 2 Report

Comments and Suggestions for Authors

The authors have improved the clarity of their work with the changes incorporated into the document. In addition, they have added the comments and suggestions made by the reviewers. However, the limitations inherent in the study design are relevant and may affect the publication of the article (although the authors provide a thorough and well-written argumentation).

Author Response

Reviewer 2: The authors have improved the clarity of their work with the changes incorporated into the document. In addition, they have added the comments and suggestions made by the reviewers. However, the limitations inherent in the study design are relevant and may affect the publication of the article (although the authors provide a thorough and well-written argumentation).

Response: Thank you for this comment and highlighting that we have now improved the clarity of our work and made the necessary suggestions that were made. We understand the pee-reviewers concerns over the limitations of the study. However, as highlighted by the reviewer, we have provided a thorough justification for choices made in this study and discussed particular limitations of the study that exist. We hope that by doing so, we have provided the justification needed for this manuscript to proceed to publication. 

Reviewer 3 Report

Comments and Suggestions for Authors

The authors responded well to the reviewer's concerns; no more questions need to be raised. The reviewer thought it had reached the requirement for published authors' work.

Author Response

Reviewer 3: The authors responded well to the reviewer's concerns; no more questions need to be raised. The reviewer thought it had reached the requirement for published authors' work.

Response: Thank you for this comment and recommending that it now reaches the requirement for publication.